

# Improving plant miRNA-target prediction with self-supervised k-mer embedding and spectral graph convolutional neural network

Weihan Zhang[1,2], Ping Zhang[3], Weicheng Sun[3], Jinsheng Xu[3], Liao Liao[1,2], Yunpeng Cao[1,2] and Yuepeng Han[1,2]

[1] CAS Key Laboratory of Plant Germplasm Enhancement and Specialty Agriculture, Wuhan Botanical Garden, The Innovative Academy of Seed Design of Chinese Academy of Sciences, Wuhan, Hubei Province, China
[2] Sino-African Joint Research Center, Chinese Academy of Sciences, Wuhan, Hubei Province, China
[3] College of Informatics, Huazhong Agricultural University, Wuhan, Hubei Province, China

Corresponding authors
Yunpeng Cao, xfcypeng@126.com
Yuepeng Han, wavekk@163.com

## ABSTRACT

Deciphering the targets of microRNAs (miRNAs) in plants is crucial for comprehending their function and the variation in phenotype that they cause. As the highly cell-specific nature of miRNA regulation, recent computational approaches usually utilize expression data to identify the most physiologically relevant targets. Although these methods are effective, they typically require a large sample size and high-depth sequencing to detect potential miRNA-target pairs, thereby limiting their applicability in improving plant breeding. In this study, we propose a novel miRNA-target prediction framework named kmerPMTF (k-mer-based prediction framework for plant miRNA-target). Our framework effectively extracts the latent semantic embeddings of sequences by utilizing k-mer splitting and a deep self-supervised neural network. We construct multiple similarity networks based on k-mer embeddings and employ graph convolutional networks to derive deep representations of miRNAs and targets and calculate the probabilities of potential associations. We evaluated the performance of kmerPMTF on four typical plant datasets: *Arabidopsis thaliana*, *Oryza sativa*, *Solanum lycopersicum*, and *Prunus persica*. The results demonstrate its ability to achieve AUPRC values of 84.9%, 91.0%, 80.1%, and 82.1% in 5-fold cross-validation, respectively. Compared with several state-of-the-art existing methods, our framework achieves better performance on threshold-independent evaluation metrics. Overall, our study provides an efficient and simplified methodology for identifying plant miRNA-target associations, which will contribute to a deeper comprehension of miRNA regulatory mechanisms in plants.

## INTRODUCTION

MicroRNAs (miRNAs) are small no-coding RNA molecules, approximately 22 nucleotides in length, that significantly influence post-transcriptional gene regulation (*Du & Zamore, 2007*; *Pasquinelli, Hunter & Bracht, 2005*). They are critical regulators of developmental and physiological processes in plants, including growth regulation, stress responses, and genome integrity preservation (*Axtell & Bowman, 2008*; *Chen, 2005*; *Liu et al., 2018*; *Pagano et al., 2021*; *Yang, Xue & An, 2007*). However, a comprehensive understanding of the nuanced interplay between miRNAs and the target they regulate remains an advancing research field. This understanding gap emphasizes the importance of additional explorations to decipher the complexity of miRNA-target association (*Alexiou et al., 2009*; *Axtell & Bowman, 2008*). Greater understanding of this relation can enrich our comprehension of many molecular-level biological phenomena and foster innovative strategies for plant breeding and crop enhancement (*Liu, Li & Cairns, 2012*; *Liu & Wang, 2019*; *Ravichandran et al., 2019*; *Riolo et al., 2021*; *Singh et al., 2023*). Therefore, the quest for this knowledge promises substantial advancements in plant biology and agriculture. Understanding and interpreting the roles and functions of miRNAs in plants is a vital area of scientific research (*Pagano et al., 2021*). Unraveling the intricate interconnections involving these molecules and the process by which they induce phenotype alterations is of utmost importance in plant biology. Among various aspects of miRNAs, their unique and highly-specific regulatory nature stands out, stimulating research interest and further examination in plant biology (*Guo et al., 2019*, *2021a*). With the progression in this research field, modern computational methods have become essential tools. These methods prioritize the use of expression data, facilitating the identification and understanding of the most physiologically relevant targets, thereby offering substantial insights into the functions of miRNAs. Such insights greatly impact our understanding of various biological phenomena.

Computational methodologies serve as an efficient avenue for establishing miRNA-target pairs (*Chen et al., 2017*; *Riolo et al., 2021*). Nevertheless, practical applications of these methodologies, especially in real-world situations, often encounter challenges (*Chen et al., 2017*; *Huang et al., 2019b*; *Zhang et al., 2021b*). These challenges primarily stem from the inherent design of the methodologies, which necessitate large sample sizes and in-depth sequencing to adequately identify potential miRNA-target pairs. Although this prerequisite is essential for the validity of the research, it introduces inevitable complications to its implementation. Consequently, the identification process evolves into an increasingly resource-intensive and time-consuming task, often resulting in limitations in its applicability, particularly in areas such as advancing plant breeding (*Asefpour Vakilian, 2020*; *Fridrich, Hazan & Moran, 2019*; *Pagano et al., 2021*). These interconnecting challenges considerably impede wider applications and restrict the potential for ongoing development and contribution of this research to the understanding of miRNA functions. This, in turn, stifles the possibility of achieving pivotal investigative breakthroughs that can only be remedied through the development of strategic solutions. The sustenance and advancement of this specialized research field are contingent upon

these methodological improvements, fostering a promising future. With these refinements, the field of miRNA research can harness its full potential, paving the way for groundbreaking scientific discoveries.

Deep learning, a significant subfield of machine learning techniques, has recently become an integral tool in genomics, proving its worth in predicting precise miRNAs and target pairs (*Kuang, Zhao & Yang, 2023*; *Wang et al., 2020*). Within the vast sphere of genomic science, comprehending the sophisticated network of interactions and connections between miRNAs and their respective targets is fundamental. It is crucial to note that this interaction chiefly regulates gene expression–a foundational process vital for numerous biological occurrences (*Chen, 2005*; *Guo et al., 2019*; *Liu, Li & Cairns, 2012*; *Pasquinelli, Hunter & Bracht, 2005*). Examples include plant growth, evolution, and an organism's adaptive response to a wide array of external environmental through complex processing layers, deep learning offers a considerable benefit in refining the prediction of these pivotal miRNA-target pairs. Consequently, it supports a more in-depth exploration of the intricate mechanisms that control gene regulatory networks, emphasizing its indispensability in contemporary genomic research.

The sphere of plant genomics requires focused attention towards the utilization of deep learning methods. Considering the vast array of miRNAs discovered within the plant kingdom, each with its important regulatory roles, the accurate prediction of these pairs is crucial for understanding plant biology specifics (*Chen, 2005*). The multifaceted and pivotal functions of these miRNAs have boosted the need for elevated computational tools and techniques, thereby endorsing the application of graph neural networks (GNNs) (*Yan et al., 2022*; *Zhang et al., 2021b*). GNNs proffer distinct benefits, augmenting their capability to facilitate computations not merely on grid-like data but also on non-Euclidean structured graph data. GNNs are designed to exploit the rich geometric data in network structures, thereby providing a robust framework that allows for the integration of a node's neighborhood information to generate an informed output (*Veličković, 2023*; *Zhang et al., 2021c*; *Zhou et al., 2020*). This characteristic has proved crucial in areas where data are predominantly non-Euclidean, enabling the modeling of complex, unstructured real-world phenomena. They excel in scenarios where data irregularities prevail and have an exceptional capacity to identify and interpret complex relationship patterns between nodes in a network, making them ideally suited for modelling complex interaction patterns between miRNAs and target genes. Among various types of GNNs, graph convolutional neural networks (GCNs) have emerged as a leading paradigm, particularly their use has transformed biological sciences where the prediction of biological relationship pairs is essential (*Huang et al., 2019a*; *Kipf & Welling, 2016*; *Zhang et al., 2019*). Functioning by approximating the spectral graph convolutions into practical spatial convolutions, GCNs have paved the way to novel methods of understanding intricate biological interactions. The ability to handle topological irregularities in data makes them highly applicable in biological networks, where the accurate prediction of relationship pairs directly influences the understanding of biological processes and mechanisms (*Sun et al., 2019*). Hence, the use of GCNs shows great promise

in decoding complex biological relationships, underscoring their importance in the advancement of science and technology.

Traditional methodologies, dependent on sequence complementarity, often showcase their limitations when confronted with complex genomic structures and interactions (*Dai & Zhao, 2011*; *Jones-Rhoades & Bartel, 2004*; *Meng, Shao & Chen, 2011*), such as psRNATarget (*Dai & Zhao, 2011*), Targetfinder (*Fahlgren et al., 2007*) and Target-align (*Xie & Zhang, 2010*). These techniques frequently struggle to appropriately manage the intricate structures, hindering advances in knowledge and technology pertaining to genomics. Hence, the amalgamation of deep learning principles and graph neural networks (GNNs) emerges as a powerful tool to tackle these challenges, such as MiRTDL based on neural network design (*Cheng et al., 2015*), SG-LSTM-FRAME based on random walk strategy (*Xie et al., 2020*) and MeSHHeading2vec framework based on graph embedding algorithms (*Guo et al., 2021b*). These approaches, underpinned by artificial intelligence, hold immense potential for critical breakthroughs within the realm of genomics. By efficaciously deciphering and predicting the myriad of interaction patterns and information flows within miRNA-mediated gene regulatory networks, our comprehension of these networks can be significantly enhanced (*Madhumita & Paul, 2022*). This enhanced understanding is particularly crucial within plant biology, where such networks often demonstrate unique complexities and potential applications (*Lai, Wolkenhauer & Vera, 2016*; *Pio et al., 2014*). Utilizing these advanced computational strategies not only enables innovative exploration and sophisticated interpretation of genomic data, but also potentiates the development of novel applications of genomic science. This paradigm shift could fundamentally transform the field, triggering a new era of discoveries and advancements (*Kurubanjerdjit et al., 2013*). Such paradigm shift can propel us further on the path to understanding life's blueprint, facilitating the development of interventions for various plant diseases and mutations, and contributing to a more sustainable and resilient agricultural sector.

In this study we introduce kmerPMTF, an innovative and intricate framework developed specifically for predicting plant-specific miRNA-target associations. The kmerPMTF combines the utilization of k-mer splitting and a high-end self-supervised neural network to astutely extract the cryptic yet crucial semantic embeddings in sequences. Through engaging k-mer embeddings, we enhance our design by constructing multiple similarity networks to designate a complex web of interrelations among the variables. By integrating graph convolutional networks, our computational model becomes more refined, extracting both profound and complex representations of miRNAs and their corresponding targets. A comprehensive evaluation performed on kmerPMTF across four unique plant data sets revealed excellent predictive performance on threshold-independent evaluation metrics, thereby situating kmerPMTF as a leading framework compared to existing methodologies. Our study emphasizes the proficiency of kmerPMTF in foreseeing new miRNA-target associations, supported by the successful application of small RNA sequencing - a process further validated by our framework. In its essence, kmerPMTF serves as a beacon of advancement, streamlining the identification of plant miRNA-target

associations. Its impact extends beyond just predictions, facilitating a deeper understanding of miRNA regulatory mechanisms within the intricate realm of plant life.

# MATERIALS AND METHODS

In the current study, we primarily aim to predict potential associations between plant miRNA and genes (MGA), treating this task as link prediction within the complex structure of a heterogeneous graph. To increase the effectiveness of our task, we propose a novel framework centered on GCN principles. This framework enables the learning of latent node representations from graph-structured data, offering a deeper understanding of the relationships within the graph. In contrast to less thorough learning methodologies, our framework emphasizes extracting significant insights captured within the relational web of a graph. In the following section, we will meticulously dissect and explain our proposed framework, offering a rigorous walkthrough and detailed exposition of its unique components. This strategy aims to enhance understanding and provide insights into our novel approach.

## K-mer frequency counter

DNA/RNA sequences, being the building blocks of genetics, demonstrate discernible structures and patterns due to their biochemical composition. This consistency makes tools derived from information theory, such as cross-entropy, suitable for performing frequency analysis on these sequences (*Compeau, Pevzner & Tesler, 2011*). Many of the methods used in this analysis break down each individual sequence into numerical components. These components often include frequencies that stem from the incidence of word types or substrings of a definite length (k-mers) within the sequences. The frequency analysis of k-mers is a valuable technique in genomics as it enables the characterization of DNA/RNA sequences into quantifiable units. If two sequences display high degrees of similarities, their respective distributions of derived k-mer frequencies also align; they echo these similarities, subsequently creating a correlation signature. In contrast, sequences that are dissimilar or unrelated will showcase contrasting frequency distributions. Recognizing this principle, we employ a counter to meticulously transform miRNA and target gene sequence information into numerical descriptors based on k-mer frequencies, thereby converting bio-information into digestible mathematical representation. Specifically, we employ an iterative fragment selection approach for each sequence, treating it as an assembly of length k. A sequence of length m, for instance, can be disassembled into m-k+1 sequence fragments, akin to a puzzle being separated into smaller pieces. The miRNA sequence comprises four bases that are fundamental to the genetic code: adenine (A), uracil (U), cytosine (C), and guanine (G). For example, in a scenario where we select a k-value of 7 during an experiment, this allows us to generate sequence fragments akin to AAAAAAA, AAAAAAU, AAAAAAC, AAAAAAG, …, and GGGGGGG, thus yielding an extensive palette of 16,384 (4^7) unique combinations. Such variations demonstrate the richness of genetic data. Consequently, the process transforms the character sequences of miRNA and target genes into a structured, standardized numerical matrix with $N \times 16,384$,

where $N$ is the total number of miRNA and target, providing a format that is highly advantageous for further computational analysis and machine learning algorithms.

## K-mer frequency counter

Embeddings are known to represent the underlying characteristics of semantics, enabling convolutional neural models to effectively capture hidden deep semantics. In the field of genomics, it has been demonstrated that k-mer embeddings offer superior performance and advantages for deep learning models (*Fang, Deng & Li, 2022*; *Trabelsi, Chaabane & Ben-Hur, 2019*). Motivated by these findings, we have designed a self-supervised deep neural network (SDNN) comprising four convolution layers, aimed at extracting deep semantics from the k-mer frequency matrix. In our approach, the frequency vector of each sequence serves as the input for SDNN and undergoes deep self-encoding and self-decoding, a process that is iterated multiple times in a loop. To facilitate learning in each fully connected layer, we apply a hyperbolic tangent activation function, which effectively centers the data around zero and enhances the subsequent layer's learning process. The final output of embedding learner is a matrix $M_{kmer} \in \mathbb{R}^{\left(N_{miRNA}+N_{target}\right) \times N_{embedding}}$, where $N_{miRNA}$ and $N_{target}$ is the number of miRNA and target gene, respectively. $N_{embedding}$ is the dimensions of learning embeddings.

## Heterogeneous graph construction

To represent known associations between miRNA and target genes more effectively, a heterogeneous graph was constructed based on miRNA-target associations, target similarity network and miRNA similarity network. Let the heterogeneous is $G = (v, \varepsilon)$, where $v = \left(N_{miRNA}, N_{target}\right)$ representing $N_{miRNA}$ miRNA nodes and $N_{target}$ target gene nodes and $\varepsilon$ is a set of edges between nodes. Suppose that the labels of some links, $\varepsilon$ in $G$ are given, the goal is to predict if there is any potential link between any miRNA-target pair that have not yet previously been established.

Here, we denote miRNA-target association as a binary matrix $A^{miRNA-target} \in \{0,1\}^{N_{miRNA} \times N_{target}}$. $A_{i,j}^{miRNA-target}$ is equal to 1, if a miRNA $m_i$ has association with a target $\ell_j$; otherwise $A_{i,j}^{miRNA-target} = 0$. The pairwise similarities between $N_{miRNA}$ miRNA are denoted as a similarity matrix $S^{miRNA}$ with $S_{i,j}^{miRNA}$ as its $(i,j)$th entry; the pairwise similarities between $N_{target}$ target genes are denoted as a similarity matrix $S^{target}$ with $S_{i,j}^{target}$ as its $(i,j)$th entry. The similarities in $S^{miRNA}$ or $S^{target}$ can be measured based on the k-mer frequency matrix by cosine or Jaccard based on a general threshold 0.5 (*Cheng et al., 2015*; *Guo et al., 2021b*; *Xie et al., 2020*). Mathematically, the heterogeneous graph $G$ can be represented by an adjacency matrix $A_H$:

$$A_H = \begin{bmatrix} S^{miRNA} & A^{miRNA-target} \\ A^{miRNA-target \, T} & S^{target} \end{bmatrix} \in \mathbb{R}^{\left(N_{miRNA}+N_{target}\right) \times \left(N_{miRNA}+N_{target}\right)}. \tag{1}$$

## Graph convolutional network

GCN (*Kipf & Welling, 2016*) and its variants (*Defferrard, Bresson & Vandergheynst, 2016*; *Du et al., 2017*; *Hamilton, Ying & Leskovec, 2017*; *Veličković et al., 2017*) generally learn

node features through a three-step process: message passing, aggregation and representation update. The critical step is feature aggregation, in which a node aggregates feature information from its topology neighbors and itself in each convolution layer. The GCN model relies on the adjacent matrix of graph and the feature matrix of nodes. For a given miRNA-target heterogeneous graph $G = A_H$, each node in a GCN network generally contains its own features that belong to the feature matrix $M_{kmer}$ (*Kipf & Welling, 2016*), so an identity matrix is always added to the adjacency matrix:

$$\mathbb{A}_H = A_H + I_H \tag{2}$$

where $A_H$ is the adjacency matrix of heterogeneous graph as Formula (1), $I_H$ is the identity matrix.

Current approaches to designing localized convolutional filters on graphs can be roughly classified into two categories: spatial and spectral approaches. Spatial-based approaches construct the filter localization based on local information from neighboring nodes, which may be limited in terms of matching local neighbors (*Bruna et al., 2013*). On the other hand, spectral-based approaches rely on the spectrum of the graph Laplacian for filter design (*Kipf & Welling, 2016*). A well-defined localization operator on graphs was introduced by employing a Kronecker delta function implemented in the spectral domain. In contrast to spatial-based approaches, spectral-based approaches typically exhibit superior performance in graph learning (*Ayat et al., 2019*; *Ding et al., 2021*; *Huang et al., 2019a*). As the number of nodes and dimensions of feature vectors in graph G usually thousands or tens of thousands. It is time consuming for traditional spatial approach when decomposing the Laplacian matrix spectrum, and convolutional operation often directly performs on global that weakens the local information (*Bruna et al., 2013*). Therefore, a spectral approach that rely on the spectrum of graph Laplacian was proposed (*Defferrard, Bresson & Vandergheynst, 2016*). Unlike the traditional spatial approach, the graph convolution in spectral approach is defined on graph as the product of the input signal and the filter $g_\theta$ in the Fourier domain. Let us denote the symmetric normalized Laplacian matrix of $A_H$ is $L_{spectral}$:

$$L_{spectral} = U_H \Lambda_H U_H^t \tag{3}$$

where $U_H$ is the eigenvector matrix and $\Lambda_H = \mathrm{diag}(\lambda_1, \ \lambda_2, \ \lambda_3, \ldots, \ \lambda_{N_{miRNA}+N_{target}})$ is the diagonal matrix of eigenvalues.

Here, the Fourier transform of $M_{kmer}$ is represented as $U_H^t M_{kmer}$. However, that calculation of getting eigenvector matrix and eigenvalues diagonal matrix is expensive with the increasing scale of graph. Thus, a modified GCN based on Chebyshev polynomials $T_K(x) = 2x T_{K-1}(x) - T_{K-2}(x)$ was used here for features representation to reduce the computational complexity (*Defferrard, Bresson & Vandergheynst, 2016*). As a result, the filter $g_\theta$ can be defined and represented as follows:

$$g_\theta(\Lambda_H) = \sum_{K=0}^{K} \theta_K T_K(\tilde{\Lambda}_H) \tag{4}$$

$$g_\theta * M_{kmer} = \sum_{K=0}^{K} \theta_K T_K(\tilde{L}_{spectral}) M_{kmer} \tag{5}$$

where $\theta \in \mathbb{R}^K$ is a vector of Chebyshev coefficients, $\tilde{\Lambda}_H = \dfrac{2\Lambda_H}{\lambda_{max}} - I_H$, $\tilde{L}_H = \dfrac{2L_H}{\lambda_{max}} - I_H$, $I_H$ is the identity matrix and K is the $K^{th}$-order neighborhood.

Since the Chebyshev polynomials are recursively (*Hammond, Vandergheynst & Gribonval, 2011*), the formulation can be simplified by limiting K = 1 (*Kipf & Welling, 2016*). Finally, the spectral GCN can be represented as:

$$g_\theta * M_{kmer} = CELU\left(\theta\left(D_H^{-\frac{1}{2}}(I_N + A_H)D_H^{-\frac{1}{2}}\right)\right) \tag{6}$$

$$\begin{bmatrix} H_{spectral}^{miRNA} \\ H_{spectral}^{target} \end{bmatrix} = g_\theta * M_{kmer} \tag{7}$$

where $D_H$ is the diagonal matrix with diagonal entry $[D_H]_{i,j} = \sum_j [A_H]_{i,j}$, $H_{spectral}^{miRNA}$ is the embedding of miRNA and $H_{spectral}^{target}$ is the embedding of target genes.

## Edge representation construction and probabilities prediction

After GCN encoding, we concatenate $H^{miRNA}$ and $H^{target}$ as the feature vector for each miRNA-target pairs, so as a result, the features of each miRNA-target pairs are the combination of the features of its miRNA and target gene. Then, the multilayer perceptron (MLP) is used as a supervised learning model to predict the association probabilities. The MLP contains three fully connected layers. The input for MLP is miRNA-target pair feature vector, which is extracted using the GCN. Since the prediction can be deemed as a two-class problem, we chose a sigmoid activation function for the final layer.

## Overall loss and optimization

For model training, we used the binary cross-entropy (BCE) loss function and the Adam optimizer for learning model parameters. The BCE $\mathcal{B}$ can be represented as follow:

$$\mathcal{B} = -\frac{\sum_{i,j \in \mathcal{Y} \cup \mathcal{Y}^-}\left(y_{ij} \log \hat{y}_{ij} + \left(\infty - y_{ij}\right)\log\left(\infty - \hat{y}_{ij}\right)\right)}{N} \tag{8}$$

where N is the number of pairs, $y_{ij}$ is the true label of the edges, which will be 1 or 0, $\mathcal{Y}$ and $\mathcal{Y}^-$ denote the set of all nodes contained in the positive edges set and negative edges set, respectively.

To mitigate the issue of overfitting, we employ $L_2$-regularization:

$$L_2 = \frac{\lambda}{2N}\sum_\omega \omega^2 \tag{9}$$

where $\lambda$ is a hyperparameter, $\omega$ is an element in the parameter matrices W.

As a result, the overall loss function for training is $\mathcal{L} = \mathcal{B} + L_2$. The whole model *via* back propagation algorithm in an end-to-end manner can be trained.

## Evaluation metrics

After constructing the framework, the final step involves evaluating the performance of our model to provide users with an understanding of its utility and weaknesses. Model evaluation is conducted using five-fold cross-validation (5-CV). We categorize all recognized miRNA-target pairs as the positive dataset and evenly divide them into five portions. An unknown miRNA-target pair suggests that no evidence from biological experimentation can confirm the association between the miRNA and target nodes. All such unspecified miRNA-target associations constitute the negative dataset. For each iteration, we select four-fifths of the positive samples and an equivalent number of randomly chosen negative examples as the training set. In contrast, the test set comprises the remainder of the positive dataset and all the remaining negative samples (*Su et al., 2022*; *Xuan et al., 2022*). The accuracy of the model is evaluated using the receiver operating characteristic curve (ROC). As a primary evaluation metric, the area under the ROC curve (AUROC) is adopted. Considering its bias towards imbalanced datasets, the precision-recall curve (PRC) is also utilized, with the area under the PRC curve (AUPRC) chosen as another primary evaluation metric. The performance of our model can be assessed using AUROC or AUPRC without requiring specific thresholds. Additionally, other evaluation metrics including accuracy (ACC), recall (REC), precision (PRE), and F1-score (F1) are calculated as follows:

$$ACC = \frac{TP + TN}{TP + TN + FP + FN} \tag{10}$$

$$REC = \frac{TP}{TP + FN} \tag{11}$$

$$PRE = \frac{TP}{TP + FN} \tag{12}$$

$$F1 = \frac{2 \times REC \times PRE}{REC + PRE} \tag{13}$$

where TP is true positive, FP is false positive, FN is false negative, and TN is true negative in the predicting results.

## Hyperparameters and model implementation

Prediction performance is contingent on several hyperparameters. The efficacy of model training is particularly influenced by two critical hyperparameters. We conducted a series of combinatorial experimental tests. The first is the learning rate of the optimizer $lr \in \{0.00001, 0.00005, 0.0001, 0.00015, 0.00003\}$, which essentially determines the step size at each iteration while moving toward a minimum of a loss function. The second is the number of epochs $ne \in \{500, 1,000, 2,000, 3,000, 5,000\}$, dictating the number of times the learning algorithm will work through the entire training dataset. Our study

involved iteratively testing on four distinct plant datasets. Our results revealed that a learning rate of 0.0001 is optimal, maintaining a delicate balance between adequate learning speed and prevention of any overshoots. Consequently, in our model, we adhered to this learning rate for training. Simultaneously, we found it beneficial to set the number of epochs to 3000, giving the model ample opportunity to learn from the data without overfitting. As for the L2-regularization coefficient $L \in \{0.0001, 0.0005, 0.001, 0.002, 0.004\}$, a parameter used to prevent overfitting by discouraging complex models, we set it to 0.0005. During the optimization process, priority was given to one specific parameter for adjustments, while the other two parameters were held constant at their nominal values. Following each optimization cycle, the optimal value of the parameter was used to redefine its nominal value.

Our model, kmerPMTF, was realized using the robust capabilities of the PyTorch framework (v2.0.0) (*Paszke et al., 2019*), built specifically for high-performance machine learning. Additional functionality was incorporated from numpy (v1.23.5), pandas (v1.5.3). Other key resources included the PyTorch Geometric (v2.3.0) (*Fey & Lenssen, 2019*), a geometric deep learning extension library for PyTorch, and scikit-learn (v1.2.2) (*Pedregosa et al., 2011*), a free software machine learning library for Python. Both the training and testing stages were performed using CUDA v11.0 on Tesla V100S, embracing the power of GPU acceleration to train our model and perform predictions efficiently.

# RESULTS

## Design of the kmerPMTF framework

The kmerPMTF is an open-source python command-line utility that serves as a fast and reliable framework for predicting plant miRNA-target associations (MTA). It utilizes the deep learning graph convolution theory to train a specific model for a given plant. The kmerPMTF framework consists of two modules: a 'graph constructing' module, which constructs a miRNA-target graph from the sequences of specific plant miRNAs and target genes, and a 'graph training' module, which utilizes a graph convolutional network to learn the representations and train a specific plant prediction model (Fig. 1). The main reason for having both modules is that distinct graph construction approach probably contributes to the graph training module. Besides, by adjusting the training parameters according to different datasets, an optimal prediction model for specific plants can be obtained. Therefore, splitting the framework into two modules can enhance the ease of use and the degree of customization. This will allow users to easily train suitable models for different plants using the kmerPMTF framework.

The 'graph constructing' module consists of two components: a k-mer frequency counter, which counts the frequency of k-mer, and a learner, which learns embeddings. This module takes the nucleotide sequences of both miRNA and target gene as input. The k-mer frequency is utilized to compute the similarity that is necessary for constructing a heterogeneous graph and feeding it to a self-supervised deep neural network (SDNN) that learns embeddings serve as attributes for the graph nodes. Next, integrate the miRNA similarity, target gene similarity, known miRNA-target pairs, and k-mer frequency as a heterogeneous graph to generate an output that can be used for graph convolution
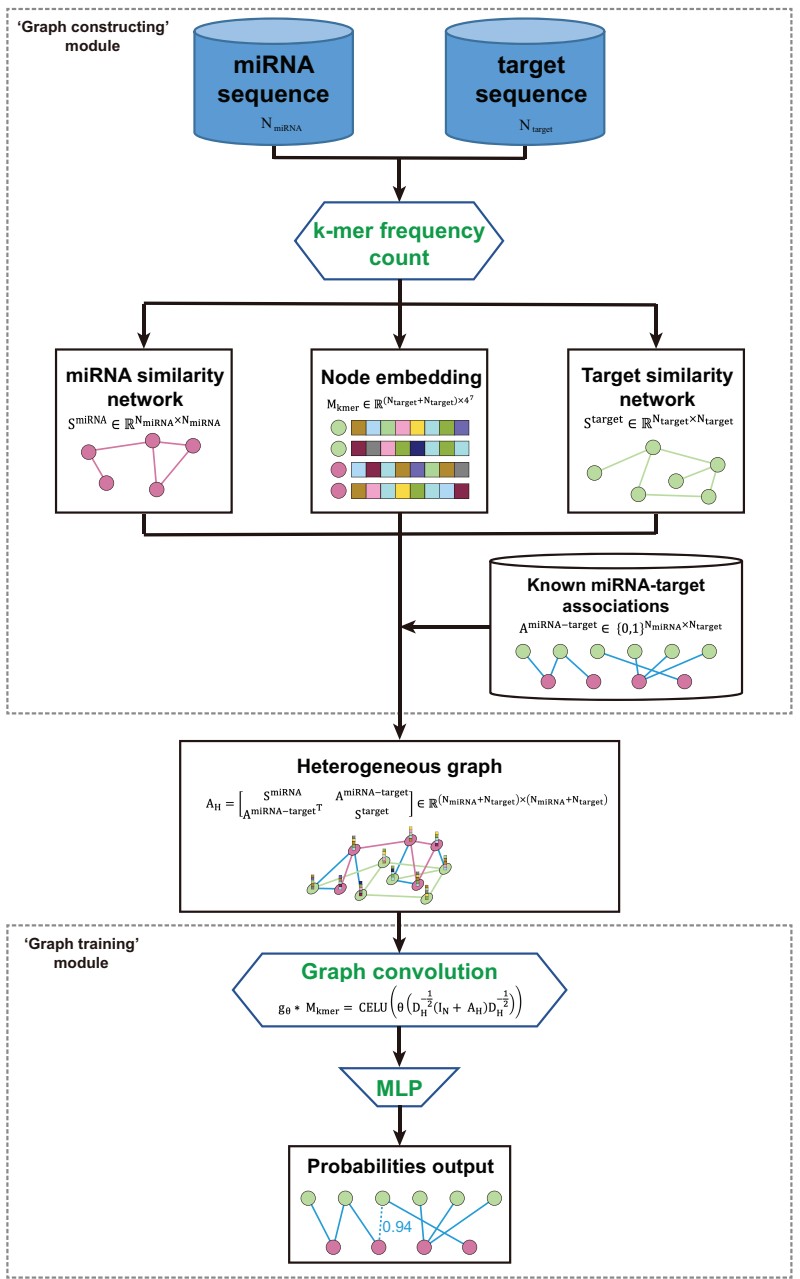

**Figure 1 Schematics of the KmerPMTF workflow.**

operations (Fig. S1A). The 'graph training' module consists of a series of graph neural network layers that comprise a typical graph convolutional network (GCN) architecture. The purpose of this architecture is to learn the topological structure and node semantics from the graph. The heterogeneous graph constructed by the previous module serves as input to this module. After extracting node representation through three layers of graph convolution, we pair the nodes to obtain edge representations, which are then fed into a multilayer perceptron (MLP) to compute the probabilities of all edges in the heterogeneous graph. We consider the prediction of associations between miRNA and target gene as a

**Table 1 Four plant datasets used in this study.**

| Species | Gene number | MIR number | Pairs |
|---|---|---|---|
| *Arabidopsis thaliana* | 27,416 | 221 | 3,646 |
| *Oryza sativa* | 42,189 | 699 | 17,809 |
| *Solanum lycopersicum* | 34,658 | 324 | 2,181 |
| *Prunus persica* | 26,873 | 293 | 6,251 |

link prediction task. The probabilities of the predicted edges between miRNA and target gene are regarded as the probability of a regulatory relationship between a specific miRNA and gene (Fig. S1B).

## The kmerPMTF outperforms published methods

The primary goal of kmerPMTF is to predict the binary classification of miRNA-target gene. We evaluated the effectiveness and performance of kmerPMTF by comparing it with popular deep learning and traditional machine learning methods. Four typical plant datasets, namely *Arabidopsis thaliana*, *Oryza sativa*, *Solanum lycopersicum*, and *Prunus persica*, were employed in the comparison. *Arabidopsis thaliana* is a classic model plant which has been widely used in plant genetic studies (*Koornneef & Meinke, 2010*; *Somerville & Koornneef, 2002*), the other three are representative model plants in the field of crops, vegetables and fruit trees (*Aranzana et al., 2019*; *Izawa & Shimamoto, 1996*; *Kimura & Sinha, 2008*), respectively. All the miRNA information and known miRNA-target gene pairs are obtained from Plant miRNA Encyclopedia project (PmiREN) (*Guo et al., 2021a*), which provides high-confidence MIR and regulatory relationships by integrating several databases and many datasets. The four plant genome datasets used in our study contained a range of 27,416 to 42,189 genes and 221 to 699 MIRs. The number of pairs formed between miRNA and their target genes ranged from 2,181 to 17,809 (Table 1).

For the deep learning comparison, we selected MiRTDL (*Cheng et al., 2015*), dgMDL (*Luo et al., 2019*) and SG-LSTM-FRAME (*Xie et al., 2020*). MiRTDL utilizes the traditional convolutional neural network (CNN) model, whereas dgMDL employs the multimodal deep belief network (DBN), which can be regarded as a stack of restricted Boltzmann machine. In contrast, SG-LSTM-FRAME is designed based on a random walk strategy to predict potential relationships. Furthermore, to demonstrate the efficacy of graph convolutional network in extracting deep topological semantics, we conducted experiments using our datasets in the MeSHHeading2vec framework (*Guo et al., 2021b*) and graph attention networks (GAT) (*Veličković et al., 2017*), which employs graph embedding and graph neural networks. We configured the parameters of the aforementioned methods in accordance with the recommended guidelines of the author. All four of these methods are categorized under deep learning. We tested our plant datasets using these models. Besides, to exemplify the effectiveness of deep learning, we conducted experiments on our dataset using traditional machine learning techniques. Specifically, we applied the Adaboost, SVM-SVC and random forest algorithms to our data. All the results
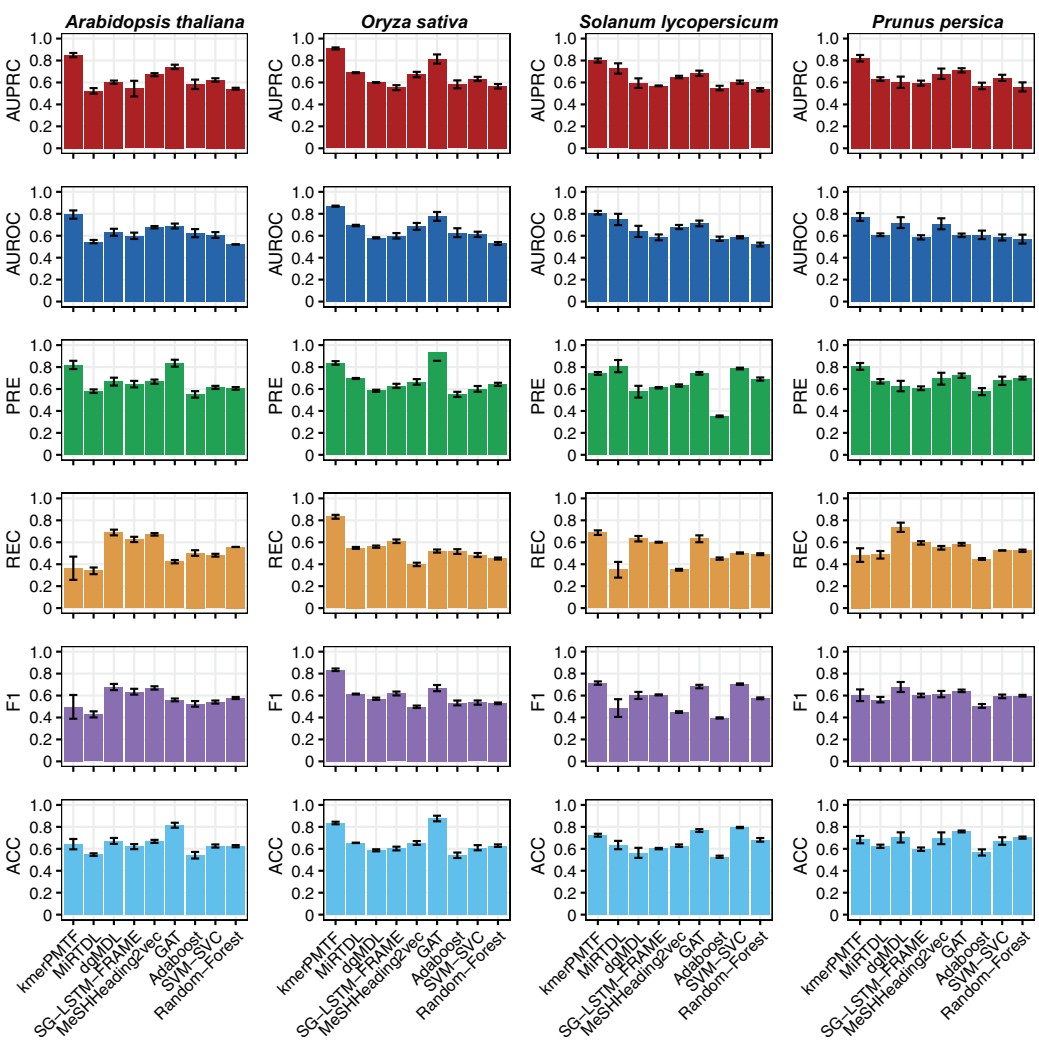

**Figure 2** The performances of kmerPMTF compare with other methods in terms of AUPRC, AUROC, PRE, REC, F1 and ACC, where error bar represents the 0.95 confidence interval of different metrics.

consistently indicate that kmerPMTF is the optimal choice for the plant datasets, as it achieves the highest AUPRC values of 0.84, 0.91, 0.80 and 0.82 in *Arabidopsis thaliana*, *Oryza sativa*, *Solanum lycopersicum* and *Prunus persica*, respectively (Fig. 2 and Fig. S2).

## Parametric sensitivity analysis

As different parametric can affect model performance, we explored the parameters of graph construction, embedding generation and model training of our experiments. We first explored the influence of the length of k-mer selected for the frequency counter. For four models in this work, we set the k-mer length to be 7 bp. This value is within the range of typical transcription factor binding site lengths (5–9 bp) (*Stormo, 2000*; *Tsai et al., 2006*; *Yu, Lin & Li, 2016*). To test the robustness of our framework, we further set a series of k-mer lengths (5, 7, 9 bp) within this range for testing. The results showed that for the four plants selected in this study, although the model performed best when the k-mer

length was 7 bp, the influence of different k-mer lengths on the fluctuations of the models AUPRC and AUROC did not exceed 5% and 8%, respectively. Secondly, we explored different methods of generating similarity network. Cosine and Jaccard similarity were tested, and we discovered that cosine similarity might be more suitable in our cases. Thirdly, we tested k-mer embedding dimensions used as node attributes for graph convolution. We selected different dimensions from the range of 16–512 for testing. The results show that when the dimension is greater than or equal to 128, the performance is improved by 2–5%, while the RAM consumption is increased by three times compared when it is less than 128. Our experiments demonstrate that selecting the appropriate hyperparameter k-mer can enhance the model effect. Additionally, our framework mitigates the impact of inaccurate k-mer selection on performance reduction.

## Examination of the graph components

Our framework, kmerPMTF, is constructed with its foundation firmly anchored on the established principles of the ChebNet graph convolutional model. kmerPMTF skillfully leverages the nuanced semantics of sequence k-mer, forming a comprehensive heterogeneous graph. This, in turn, creates a rich tapestry of information, ripe for analysis by our model. Further adding a level of sophistication, the framework employs a spectral graph convolutional network. This method facilitates the extraction of profound and detailed representations, which prove instrumental for accurate and precise predictions. As part of our rigorous validity checks and to assess the robustness of kmerPMTF, we adopt an experimental approach during the training phase. Certain components from the heterogeneous graph are deliberately omitted in this process. The objective of this strategy is to zero in on the individual contributions of the elements and test the framework's capability to sustain its performance amidst these variable conditions. In parallel, we also design and implement a node feature elimination strategy. This systematic approach is aimed at assessing the degree to which each k-mer embedding is necessary and contributes to the overall success of the model. We initiate five model variations as follows, each evolved from the original and expressing a unique set of parameters and settings.

– miT+TT+mimi: It is a Chebyshev GCN model that uses a graph with all elements but lacks node features.
– miT+TT+feat: It is a Chebyshev GCN model that uses a graph including node features but lacks a miRNA-miRNA semantics similarity network.
– miT+TT: It is a Chebyshev GCN model that uses a graph which lacks both node features and miRNA-miRNA semantics similarity network.
– miT+mimi+feat: It is a Chebyshev GCN model that uses a graph including node features but lacks a target-target semantics similarity network.
– miT+mimi: It is a Chebyshev GCN model that uses a graph which lacks both node features and target-target semantics similarity network.

Upon employing a 5-fold cross-validation for each variant model, a significant decrease in the AUROC is observably evident when compared with the original model

**Table 2 AUROC of different variation model in ablation study.**

| Models | Arabidopsis thaliana | Oryza sativa | Solanum lycopersicum | Prunus persica |
|---|---|---|---|---|
| miT+TT+mimi | 0.84 | 0.91 | 0.80 | 0.82 |
| miT+TT+feat | 0.58 | 0.61 | 0.54 | 0.55 |
| miT+TT | 0.51 | 0.52 | 0.51 | 0.49 |
| miT+mimi+feat | 0.64 | 0.53 | 0.57 | 0.66 |
| miT+mimi | 0.59 | 0.71 | 0.61 | 0.61 |

(miT+TT+mimi) (Table 2). The integration of features from miRNA and target genes markedly improves the model's capability to assimilate multiple strata of representation, thereby potentially amplifying its performance.

## Case studies

In this section, we conduct case studies to further validate the predictive performance of the model kmerPMTF in real situations. In peach, the gene *PpTIR1* (*Prupe.4G037400* and *Prupe.4G037200*) as a target of miR393a and miR393b and have been verified using the GUS assay (*Ma et al., 2023*). We also select miR171c and miR408 in *Arabidopsis* as a case, which is involved in plant growth and development, stress response and hormone signaling (*Gao et al., 2022*; *Pei et al., 2023*). As biologists are more interested in the top prediction, we finally choose the top five associated target genes from the prediction results and validate them on the latest database PmiREN (*Guo et al., 2021a*) and TarDB (*Liu et al., 2021*) (Table 3). We can find that all the top five target genes have been supported by existing databases and experiment study. The results from the case studies suggest that kmerPMTF is an effective tool in plant miRNA-target pair prediction.

## DISCUSSION

In this study, we introduce a novel framework named kmerPMTF (k-mer-based prediction framework for plant miRNA-target) for predicting the targets of microRNA (miRNA) in plants. Current computational approaches, which rely on large sample size and high-depth sequencing, have limitations in their applicability in plant breeding. The kmerPMTF framework overcomes these constraints by using k-mer splitting and a deep self-supervised neural network for efficient extraction of latent semantic embeddings of sequences. The framework creates multiple similarity networks based on k-mer embeddings and applies graph convolutional networks to develop deep representations of miRNAs and targets, and to calculate their potential association probabilities.

The model's performance was evaluated on four typical datasets: *Arabidopsis thaliana*, *Oryza sativa*, *Solanum lycopersicum*, and *Prunus persica*, where it achieved AUPRC values of 84.9%, 91.0%, 80.1%, and 82.1% respectively in a 5-fold cross-validation. These results indicate a superior level of performance on threshold-independent metrics when compared to several other state-of-the-art methods, demonstrating the effectiveness of kmerPMTF. Moreover, our model serves as a guiding reference for interaction prediction research across numerous computational biology domains, especially offering substantial

**Table 3 The top 5 predicted target genes of four real cases.**

| Species | miRNA | Ranking | Target | Score | Evidence |
|---|---|---|---|---|---|
| Peach | miR393a | 1 | Prupe.4G187100 | 0.95 | PmiREN |
| | | 2 | Prupe.3G311800 | 0.93 | PmiREN |
| | | 3 | Prupe.4G037400 | 0.92 | GUS assay |
| | | 4 | Prupe.4G037200 | 0.91 | GUS assay |
| | | 5 | Prupe.8G042000 | 0.91 | PmiREN |
| Peach | miR393b | 1 | Prupe.3G311800 | 0.98 | PmiREN |
| | | 2 | Prupe.4G037400 | 0.98 | GUS assay |
| | | 3 | Prupe.1G173300 | 0.97 | PmiREN |
| | | 4 | Prupe.8G241400 | 0.96 | PmiREN |
| | | 5 | Prupe.2G295400 | 0.96 | PmiREN |
| Arabidopsis | miR171c | 1 | AT3G47170 | 0.98 | PmiREN |
| | | 2 | AT3G60630 | 0.98 | PmiREN |
| | | 3 | AT4G00150 | 0.98 | PmiREN, TarDB |
| | | 4 | AT2G45160 | 0.98 | PmiREN, TarDB |
| | | 5 | AT5G08300 | 0.97 | PmiREN, TarDB |
| Arabidopsis | miR408 | 1 | AT1G72230 | 0.99 | PmiREN |
| | | 2 | AT2G44790 | 0.99 | PmiREN |
| | | 3 | AT3G02200 | 0.98 | PmiREN, TarDB |
| | | 4 | AT1G15830 | 0.97 | PmiREN |
| | | 5 | AT2G02850 | 0.96 | PmiREN, TarDB |

insights for relative studies focused on miRNA-target interaction prediction (*Liu et al., 2020*; *Wang et al., 2022*; *Zhang et al., 2021a*, *2021b*).

Since kmerPMTF is a machine learning model, its performance heavily depends on the quality and completeness of the input data. If the data is biased, incomplete, or noisy, the performance of the model could be significantly affected. The graph convolutional networks require significant computational resources, especially for large datasets. This might limit the usage of the model in resource-constrained environments or for very large-scale analyses. Deep learning models, in general, are often criticized for being 'black boxes', as they make predictions that are hard to interpret. While kmerPMTF might be able to achieve high prediction accuracy, it might not provide direct biological insights into why certain miRNAs associate with specific targets. While these potential common limitations exist, none of them undermine the overall value of the framework. They merely point to future works that could further improve upon the existing method.

## CONCLUSIONS

In summary, the study provides a simplified and efficient method for identifying plant miRNA-target associations. On a broader scope, it contributes to a deeper understanding of miRNA regulatory prediction work in plants. The model and methodology also have a significant practical application, as they can enhance plant breeding efforts by enabling

accurate predictions about miRNA and target associations without the need for extensive sample sizes or in-depth sequencing.

## ACKNOWLEDGEMENTS

The computations in this article were run on the bioinformatics computing platform of the Public Laboratory Platform, Wuhan Botanical Garden, Chinese Academy of Sciences.

### Funding

This work was supported by the Natural Science Foundation of Hubei Province (2023AFB1036), the National Natural Science Foundation of China (32372658 and 31872059), the Sino-Africa Joint Research Center, CAS (SAJ202109), and the Knowledge Innovation Program of Wuhan Basic Research (202200040). The funders had no role in study design, data collection and analysis, decision to publish, or preparation of the manuscript.

### Grant Disclosures

The following grant information was disclosed by the authors:
Natural Science Foundation of Hubei Province: 2023AFB1036.
National Natural Science Foundation of China: 32372658 and 31872059.
Sino-Africa Joint Research Center, CAS: SAJ202109.
Knowledge Innovation Program of Wuhan Basic Research: 202200040.

### Competing Interests

Yunpeng Cao is an Academic Editor for PeerJ.

### Author Contributions

- Weihan Zhang conceived and designed the experiments, performed the experiments, analyzed the data, prepared figures and/or tables, authored or reviewed drafts of the article, and approved the final draft.
- Ping Zhang conceived and designed the experiments, performed the experiments, prepared figures and/or tables, and approved the final draft.
- Weicheng Sun conceived and designed the experiments, performed the experiments, prepared figures and/or tables, authored or reviewed drafts of the article, and approved the final draft.
- Jinsheng Xu conceived and designed the experiments, performed the experiments, prepared figures and/or tables, authored or reviewed drafts of the article, and approved the final draft.
- Liao Liao conceived and designed the experiments, performed the experiments, analyzed the data, authored or reviewed drafts of the article, and approved the final draft.
- Yunpeng Cao conceived and designed the experiments, analyzed the data, prepared figures and/or tables, and approved the final draft.

- Yuepeng Han analyzed the data, authored or reviewed drafts of the article, and approved the final draft.

## Data Availability

The data is available at figshare: Zhang, Weihan (2023). kmerPMTF-data. figshare. Dataset. https://doi.org/10.6084/m9.figshare.24916023.v1.

The code is available at GitHub and Zenodo:

- https://GitHub.com/Weihankk/kmerPMTF/tree/main
- Zhang, W. (2024). kmerPMTF. Zenodo. https://doi.org/10.5281/zenodo.10681653.

## Supplemental Information

Supplemental information for this article can be found online at http://dx.doi.org/10.7717/peerj.17396#supplemental-information.

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
