# Peer review of "Improving plant miRNA-target prediction with self-supervised k-mer embedding and spectral graph convolutional neural network"

_PeerJ, doi:10.7717/peerj.17396_

## Round 0.1 · original submission · Major Revisions

Please revise the manuscript according to the comments from the reviewers.

Reviewer 1 ·

Basic reporting

1.English expressions need to be edited more careful and more native, in this manuscript, there are some mistakes. For example, the order of the authors’ names is wrong in the original paper, family name should be put at last. The authors should revise this mistake according to authorship conventions for scientific paper.
2. The workflow in Figure 1 is unspecific and less organized. It's more like a stack of terminologies than a higher-level summary of the existing data and methodologies, please improve it.
3. Source code and data used for bioinformatics analysis should be available to test the method and validate reported results.
4.The advancement of interaction prediction research in various fields of computational biology would provide valuable insights into miRNA-target, such as miRNA-lncRNA interaction prediction. The authors should discuss it as the future direction. Important computational models in these fields should be cited. Some recommended studies are helpful (PMIDs: 33588070, 36305458, 34232474, and DOI: 10.1016/j.knosys.2019.105261).
5. The authors should carefully check and unify the information of references. Some references lack the information of volume or contain the wrong page number.
6. Future work and limitations of the proposed algorithm should be addressed in the manuscript for further research.
7. The authors use ACC, REC, PRE and F1 as evaluation metrics, but I do not find their values in ablation study and performance comparison. In fact, other conventional evaluation metrics (e.g. AUPR and Mcc) should be provided to further evaluate the performance between their model and other methods.

Experimental design

N/A

Validity of the findings

N/A

Additional comments

N/A

Reviewer 2 ·

Basic reporting

This manuscript introduces KmerPMTF, a novel machine learning framework that utilizes Kmer embeddings, an Autoencoder framework, and a spectral GNN framework to predict miRNA-target associations. The authors state that their model achieves better performance compared with several state-of-the-art methods based on the AUPRC score in a 5CV evaluation and four typical plant datasets. The writing is also clearly written and understable.

However, before deciding to accept it, I have a few concerns or comments that need to be addressed by the authors.

General comments:
1. In introduction, although authors spend lots of paragraphs to show the important of miRNAs in plant research, the advantages of using computational methods, deep learning and GNN techniques, the significance of this work, the authors did not mention any related work or discuss the drawbacks of existing research in this area. As a reader unfamiliar with this field, I would appreciate more information on state-of-the-art (SOTA) methods and relevant studies.
2. In “Heterogeneous graph construction”, the authors state that the miRNA and target similarity graphs are constructed via the cosine and jaccard scores based on k-mer frequency matrix. They didn’t mention how to determine if two nodes are connected. For example, it is unclear whether there are any thresholds for the cosine and Jaccard scores used in the graph construction.
3. In line 294, the authors said “the prediction can be deemed as a two-class problem”. They didn’t mention anything about how to build the training data (e.g., how many true positives and true negatives? How did they generate these data? How many data are used for training, validation, and test in 5CV, etc.)
4. The authors used 4 SOTA relevant models and a few machine learnings for performance comparison. I think these models are not comprehensive. I would recommend to include a few more GNN models (e.g., GraphSage, GAT, etc.), some graph completion models (e.g, TransE, RotatE, ComplEx, DistMult, etc.), and random forest for comparison.
5. In “Evaluation metrics” section, the authors said they used multiple metrics: AUROC, AUPRC, ACC, REC, PRE, F1-score, However, in the result, the authors only show the AUPRC in 5CV. It would be helpful to show all of these metrics for a comprehensive evaluation.
6. In figure 2, instead of using bar plot, the boxplot may be a better visualization method to show all of 5 CV results because we can see its variation.
7. Line 335-341, when you separately set the parameters LR, Epochs, and L2-regularization to 0.0001, 3000, 0.0005. Did you randomly pick them? Or did you try multiple different parameter combinations? If it is latter, please illustrate how you conducted it.
8. It would be better if the authors can provide the parameter settings for all models they used for evaluation


Minor comments:
1. Line 196: “thus yielding an extensive palette of 16,384 (47) unique combinations.” -> “thus yielding a total of 16,384 (4^7) unique combinations.”
2. Lines 197-199: For the sentence “Consequently, the process transforms the character sequences of miRNA and target genes into a structured, standardized numerical matrix with 16,384 dimensions”, if I understand it correctly, the numerical matrix should be N x 16,384, which should be only 2 dimensions. The authors probably need to re-write this sentence to avoid confusion.
3. Lines 241-242: For the sentence “The GCN model relies on two input matrices, the adjacent matrix of graph and the feature matrix of every node.” -> “The GCN model relies on the adjacent matrix of graph and the feature matrix of nodes.”. Each node has an embedding and they all form a feature matrix.
4. Lines 334-335: “Our study involved iterative testing on four distinct plant datasets.” -> “Our study involved iteratively testing on four distinct plant datasets.”
5. Lines 390-392: In the sentence “We 391 evaluated the effectiveness and performance of kempt by comparing it with popular deep learning and traditional machine learning methods”, what is “kempt”?
6. Lines 505-506: “the study provides a simplified and efficient mechanism for identifying plant miRNA-target associations.” Note that this contribution of this study didn’t provide a novel biological mechanism or mathematical mechanism but rather a model. So it can’t state “provide an efficient mechanism”.
7. Lines 506-507: “it contributes to a deeper understanding of miRNA regulatory mechanisms in plants.”. As the authors mentioned in lines 498-499, this model cannot provide any explanation for “miRNA regulatory mechanism” but just a prediction between “gene-miRNA” association. Please avoid mentioning about “understanding mechanism”.

Experimental design

no comment

Validity of the findings

no comment

---

## Round 0.2 · Major Revisions

Please address the concerns raised by the reviewer #2.

Reviewer 1 ·

Basic reporting

The authors have addressed my comments and I agree acceptance.

Experimental design

The authors have addressed my comments and I agree acceptance.

Validity of the findings

The authors have addressed my comments and I agree acceptance.

Additional comments

The authors have addressed my comments and I agree acceptance.

Reviewer 2 ·

Basic reporting

Thanks to the authors for addressing most of my concerns and comments from the previous review round. However, I still have a main concern after the authors showed their evaluation results using the box plot.

According to the box plot provided in supplemental figure 2, I can see that apart from the metrics of AUROC and AUPRC, kmerPMTF is not always the best model across four evaluation plant datasets. For example, for the "ACC" metric, kmerPMTF consistently underperforms compared to GAT across all datasets. The authors didn't attempt to explain the reason, but only emphasized that kmerPMTF is the optimal model for the plant datasets because it outperforms other models in AUPRC. Additionally, their claim in the abstract, introduction, and discussion about achieving better accuracy than other state-of-the-art methods is not supported by their results.

I will only accept this manuscript if the authors can effectively address this concern.

Minor comments:
1. In lines 446 - 447, the authors stated " we conducted experiments using our datasets in the MeSHHeading2vec framework (Guo et al. 2021b) and graph attention networks (GAT) (Veličković et al. 2017), which employs graph embedding and convolutional algorithms". However, the graph attention networks (GAT) don't belong to the graph convolutional networks and thus it doesn't use convolutional algorithms. Please refer to this paper: https://computationalsocialnetworks.springeropen.com/articles/10.1186/s40649-019-0069-y.

Experimental design

no comment

Validity of the findings

no comment

---

## Round 0.3 · Major Revisions

One of the reviewers still had some concerns raised in the your manuscript, and hope that you can address these concerns.

Reviewer 2 ·

Basic reporting

I appreciate the authors' response to my concerns about the model performance, and their efforts in providing evidence to justify the use of AUPRC as main metric for model performance. However, their responses are still not adequate for addressing my concerns.

First, I agree that AUPRC is one of common and efficient metrics to evaluate machine learning models because it is independent of classification threshold. However, although it can provide a "average" performance of model across different classification thresholds, we can't ignore the importance of accuracy and F1 score in model performance. In practice, the researchers will not try comparing the results using a range of thresholds. Instead, they often use the default threshold (e.g., 0.5), which as you mentioned in your response is widely used in most relevant studies. Therefore, you can make the users believe that your model is better only when it can outperform the SOTA models in most evaluation metrics based on the widely used threshold. Most of your reference evidences (e.g., Table 3 & 4 in https://academic.oup.com/bib/article/23/1/bbab513/6456297, Table 3 in https://academic.oup.com/bib/article/23/1/bbab543/6470964) show that their models are the best in most metrics across most evaluation datasets. It is true that it is less likely that the proposed model is always better than all SOTA methods. But when demonstrating that a proposed model outperforms the relevant SOTA models, we should show its superiority most of the time under default or common settings, instead of solely relying on one or two metrics, such as AUPRC in this case.

According to the authors' response, the objective of their model is to provide a more accurate ranked list of predicted scores for users in the real case. I will suggest the authors to provide a few use cases based on the real application, as seen in most relevant studies (https://doi.org/10.1093/bib/bbac361, https://academic.oup.com/bib/article/24/1/bbac495/6918743, https://doi.org/10.1093/bib/bbad317, https://doi.org/10.1093/bib/bbaa140) they referred to in their responses, to support your responses.

Overall, if the authors can provide a few use cases (one or two) to efficiently demonstrate that the top predicted results by their model can be helpful in the real studies (probably replicate the use cases in the relevant papers as comparison), I believe that it can strongly support the superiority and efficiency of their model in identifying plant miRNA-target associations.

Experimental design

no comment

Validity of the findings

no comment

Additional comments

no comment

---

## Round 0.4 · accepted · Accept

The example of analysis was added as requested by the reviewer.